# MiRNAs as Noninvasive Biomarkers and Therapeutic Agents of Pituitary Adenomas

**DOI:** 10.3390/ijms21197287

**Published:** 2020-10-02

**Authors:** Ozal Beylerli, Narasimha M. Beeraka, Ilgiz Gareev, Valentin Pavlov, Guang Yang, Yanchao Liang, Gjumrakch Aliev

**Affiliations:** 1Central Research Laboratory, Bashkir State Medical University, 450008 Ufa, Republic of Bashkortostan, Russia; obeylerli@mail.ru (O.B.); ilgiz_gareev@mail.ru (I.G.); pavlov@bashgmu.ru (V.P.); 2Department of Biochemistry, JSS Academy of Higher Education & Research, CEMR lab, DST-FIST Supported Department and Center, Mysuru 570015, Karnataka, India; bnmurthy24@gmail.com; 3Department of Neurosurgery, the First Affiliated Harbin Medical University, Harbin 150001, China; liangyanchao@hrbmu.edu.cn; 4Institute of Brain Science, Harbin Medical University, Harbin 150001, China; 5Sechenov First Moscow State Medical University Sechenov University, 119146 Moscow, Russia; 6Research Institute of Human Morphology, Russian Academy of Medical Science, 117418 Moscow, Russia; 7Institute of Physiologically Active Compounds, Russian Academy of Sciences, Chernogolovka, 142432 Moscow, Russia; 8GALLY International Research Institute, 7733 Louis Pasteur Drive, #330, San Antonio, TX 78229, USA

**Keywords:** microRNAs, pituitary adenoma, circulating, diagnosis, therapy, exosomes

## Abstract

Pituitary adenoma (PA) accounts for 10–15% of all intracranial neoplasms. Even though most pituitary adenomas are benign, it is known that almost 35% of them exhibit an aggressive clinical course, including rapid proliferative activity and invasion of neighboring tissues. MicroRNAs (miRNAs) are short single-stranded RNA molecules that can influence post-transcriptional regulation by controlling target genes. Based on research data on miRNAs over the past 20 years, more than 60% of genes encoding human proteins are regulated by miRNAs, which ultimately control basic cellular mechanisms, including cell proliferation, differentiation, and apoptosis. Dysregulation of miRNAs has been observed in a number of diseases, especially tumors like PA. A majority of miRNAs are expressed within the cells themselves. However, the circulating miRNAs can be detected in several biological fluids of the human body. The identification of circulating miRNAs as new molecular markers may increase the ability to detect a tumor, predict the course of a disease, plan to choose suitable treatment, and diagnose at the earliest signs of impending neoplastic transformation. Therapy of PAs with aggressive behavior is a complex task. When surgery and chemotherapy fail, radiotherapy becomes the treatment of choice against PAs. Therefore, the possibility of implementing circulating miRNAs as innovative diagnostic and therapeutic agents for PA is one of the main exciting ideas.

## 1. Introduction

Pituitary adenomas (PAs) are benign, typically slowly progressing tumors of the adenohypophysis that represent 10–15% of all intracranial tumors, occurring in almost 20% of the general population. However, some of these tumors, such as invasive PAs, show a clinically aggressive behavior [1]. Invasive PAs are generally considered as those with a massive invasion of surrounding tissues, a large size, and rapid growth, which show resistance and/or multiple recurrences in spite of optimal use of standard therapies (surgery, conventional medical treatments, and radiotherapy) and, in some patients, have a fatal outcome [2,3]. MicroRNAs (miRNAs) are a class of noncoding RNAs that regulate gene expression at the post-transcriptional level. They bind to 3′-untranslated region (3′-UTR) of target mRNAs and negatively control gene translation or cause mRNA degradation [4]. It has been proven that miRNAs play a significant role in various biological processes, including the cell cycle, proliferation, differentiation, and cell apoptosis.

Most miRNAs are expressed within the cells themselves. However, biological fluids of the human body, such as blood, urine, saliva, and cerebrospinal fluid, contain numerous miRNAs, often referred to as circulating miRNAs [5]. Approximately 98% of the extracellular or circulating noncoding RNAs are miRNAs, which are reported in several gene databases as noninvasive biomarkers. (http://mirandola.iit.cnr.it/) [6]. In human biofluids, miRNAs circulate in extracellular vesicles (EVs) (apoptotic bodies, microvesicles, exosomes) and bound to macromolecular complexes, such as the Argonaute 2 (Ago 2) protein or lipoproteins (Figure 1) [6]. Exosomes are composed of a wide variety of miRNAs implicated in the pathogenesis of several diseases, including cancers. It has been proven that exosomes and the miRNA–Ago2 complex play an important role in intercellular communication. Exosomes, in addition to transferring miRNAs, are used to transfer other information, such as DNA, mRNA, proteins, etc., from one cell to another, and tumor cells play a particularly important role in exosome production [7]. Progression from benign PAs to its aggressive forms is accompanied by cumulative changes at the molecular level, where miRNAs play a direct role [8]. Dysregulation of miRNAs in tumor cells is a hallmark of a tumor, and miRNAs can play a role as tumor suppressors or oncogenes depending on the cellular context and various functions of the target genes [9]. Oncogenic miRNAs, often referred as the “oncomiRs”, can foster tumor progression by inhibiting the expression of tumor suppressor genes involved in different biological processes. Oncogenic miRNAs function as oncogenes to promote proliferation, inhibit apoptosis, induce tumor angiogenesis, and augment the oncogenic effects of transcription factors like MYC [10]. Oncogenic miRNAs, viz. miR-221 and miR-222, are reported to be involved in enhancing cell survival, consequently facilitating TRAIL resistance by downregulating p27kip1, PTEN, and TIMP3 proteins [11]. Further, these miRNAs also modulate the intrinsic pathways of apoptosis in human epithelial cells. miR-125b can target BAK1 signaling and enhance the uncontrolled proliferation of prostate and breast cancer cells [12,13]. miR-24 is reported to impair p16INK4a expression in cervical cancer cells [14]. Oncogenic miR-504 can also mitigate p53-mediated cell cycle arrest in colorectal carcinoma, U2OS osteosarcoma cells, and H460 lung carcinoma cells [15,16,17].

Thus, oncomirs can indirectly promote tumor genesis and growth. In addition, such miRNAs are mainly overexpressed in the tumor. Tumor suppressor miRNAs can decrease oncogene expression by binding to miRNA target sites in mRNA 3′-UTRs. Therefore, they can inhibit the genesis and progression of tumors. Unlike “oncomiRs”, the level of expression of such miRNAs in tumors is reduced [9]. This study is aimed at providing the latest achievements on miRNAs as biomarkers as well as therapeutic agents for PAs.

## 2. The Current State of miRNA Research in PAs

In the last decade, the aberrant expression of miRNA has been associated with human tumors, including PAs. Aberrant expression of miRNAs in PAs are often coupled with complete “chaos” in cell regulation, including uncontrolled cell proliferation, cell death modulation, tumor suppressor avoidance, tumor invasion, and angiogenesis [9]. Comparative miRNA expression profiling studies using biopsy samples from PAs have clearly indicated the differential expression of various miRNAs in PA cells compared to normal cells, emphasizing the role of miRNAs in the pathogenesis of different types of PAs [9]. However, this field of research is still relatively new, and there is much to be discovered before the role of miRNAs will definitely be proven in PA pathogenesis. For example, the ability of miRNAs to inhibit many different mRNA targets and signaling pathways at the same time may be useful for miRNAs where all targets are known. However, as miRNA appears to be cell type-specific, understanding how they function in other cell types will be critical to determine the specificity and off-target effects [9].

Some miRNAs are directly involved in the development of PA through the modulation of cell differentiation and apoptosis by targeting oncogenes and/or tumor suppressor genes. For instance, miR-410-3p has been traditionally described in gonadotroph and corticotroph PAs as an oncomiR; however, in somatotroph PAs, this miR-410-3p acts as a tumor suppressor [18]. Whether or not other miRNAs regulate PA pathogenesis by similarly targeting tumor oncogenes or tumor suppressor genes needs further study. A brief summary of differentially expressed miRNAs in PAs with their targets and implications is tabulated in Table 1 [19,20,21,22,23,24,25,26,27,28,29,30,31,32].

Research and development of effective therapeutic agents for various human diseases have long-established procedures and processes. However, these same procedures and processes are not easy to adapt for biomarker studies, especially for early diagnosis and prediction of tumors. At the end of the 20th century, the National Cancer Institute (NCI) and other research groups took the initiative to develop a systematic process for identifying and validating biomarkers. Because the detection of tumors in early stages is expected to bring great benefits to public health, the NCI has created the Early Detection Research Network (EDRN), both to identify and validate tumor biomarkers and to develop a systematic process for identifying and validating biomarkers for screening and diagnosis [33]. This EDRN scheme includes the use of a five-phase algorithm to research and develop effective biomarkers for early detection of tumors and a precancerous condition. This phased approach has been widely accepted by the biomarker community.

In recent years, significant success has been achieved in the use of biomarkers in the diagnosis of central nervous system (CNS) tumors, which are already used in everyday medical practice, such as methylation in promoter O6-methylguanine-DNA-methyltransferase (MGMT) gene in gliomas [34]. One of the most widely studied biomarkers is circulating miRNAs. Circulating miRNAs can be used as biomarkers in various tumor types, including endocrine tumors (Table 2) [35,36,37,38,39].

Although circulating miRNAs are not currently used in clinical practice, advances in this area show that their effectiveness in the diagnosis and prognosis of tumors can be crucial and replace existing difficulties in modern diagnostic practice. The significant advantages of miRNAs as molecular diagnostics are: (1) noninvasive type of detection; (2) stable and abundant miRNAs in human biological specimens; (3) circulating miRNAs are highly sensitive to pathology; (4) circulating miRNAs can be detected in the early stages of the disease, whereas protein markers are found in the blood circulation only when a significant part of the tissue damage has already occurred; (5) miRNAs play a role in almost all cellular functions; and (6) miRNAs, as information, in EVs or as part of the miRNA–Ago2 complex play a role in tumor intercellular communication [40,41].

PA is mainly diagnosed through neuroimaging techniques (magnetic resonance imaging (MRI)), hormones level, and tissue biopsy, but they all have limitations [2,3]. For instance, neuroimaging techniques can only detect established tumors with sufficient mass. Currently, the vast majority of surgical interventions, including biopsy, on the pituitary gland are carried out through the endoscopic endonasal transsphenoidal approach in its various modifications. Over the past decades, there has been a sharp decrease in the number of complications when performing such interventions. However, even with the most modern technologies, it is not always possible to avoid both mild and life-threatening complications. One of the most dangerous is intraoperative bleeding in the tumor bed [42,43]. In addition, repeated surgery and sampling in order to define the real molecular profile of tumor progression is not always possible. Liquid biopsy is a real-time sampling of biomarkers from biofluids, such as circulating miRNAs. Liquid biopsy of miRNAs could be considered as a promising approach for noninvasive detection, molecular characterization, and monitoring of the progression of aggressive PAs and CNS tumors. Early identification of PAs with aggressive behavior is challenging but is of major clinical importance as it is associated with increased morbidity and even mortality [44,45]. In this scenario, it is reasonable to assume that tumor progression will increase the expression level of certain oncogenic circulating miRNAs, for example, in the bloodstream, in patients with PAs. As a result, circulating miRNAs are becoming candidates of emerging noninvasive biomarkers.

## 3. Circulating miRNAs and PA

PAs are hormone-secreting tumors, so circulating hormones in the blood provide an excellent opportunity to control tumor growth and its function using hormonal tests. Therefore, the role of circulating miRNAs may be less important. However, with regard to nonfunctional PA (NFPA) or PA with aggressive behavior, circulating miRNAs as noninvasive blood-based biomarkers can help in the diagnosis, prognosis, and monitoring of patients during the postoperative period [46].

RNA sequencing analysis of preoperative and late postoperative plasma samples collected 3 months after pituitary surgery have exhibited differential expression of miRNAs [47]. In this study, the patients of different groups with growth hormone (GH)-secretion, follicle-stimulating hormone (FSH)/luteinizing hormone (LH), and hormone-immunonegative (HN) groups are associated with differential expression of 3, 7, and 66 circulating miRNAs, respectively [47]. Next, circulating miRNAs with higher coverage for validation were selected from significant circulating miRNAs. miR-143-3p in FSH/LH; miR-26b-5p, miR-126-5p, and miR-148b-3p in HN; and miR-150-5p in GH preoperative and postoperative samples were evaluated. Furthermore, circulating miR-143-3p was confirmed by real-time polymerase chain reaction (qRT-PCR) to be significantly upregulated in the plasma of patients with preoperative FSH/LH adenomas compared with the late postoperative plasma samples of these patients. In addition, a decrease in circulating miR-143-3p in postoperative FSH/LH samples was noticed. There was a significant reduction in the size of these tumors as well, which may have resulted in reduced circulating miR-143-3p level, indicating successful surgery. Receiver operating characteristic (ROC) curve analysis showed that the area under the ROC (AUC) of circulating miR-143-3p was 0, 79, with a sensitivity and specificity of 81.8% and 72.7%, respectively. ROC curve analysis of circulating miR-143-3p exhibited strong differentiation power between the plasma of patients with preoperative FSH/LH PAs and the late postoperative plasma samples of these patients. Their data identified circulating miR-143-3p as a potential noninvasive biomarker for patient follow-up only for FSH/LH PAs after transsphenoidal surgery, but its application for evaluating tumor recurrence needs further investigation. It has been proven that a decrease in miR-143-3p expression is observed in various tumors, which increases the likelihood of its tumor suppressor function [48,49,50]. Amaral et al. found a low expression of miR-143-3p in adrenocorticotropic hormone-secreting adenoma (ACTH)-secreting adenomas compared with normal pituitary tissue [51]. Zhang et al. showed in their study that miR-143 was suppressed in PA tissues compared with normal pituitary tissues, and overexpression of miR-143-3p inhibited cell proliferation by suppressing the K-Ras [52].

RNA sequencing study indicated 169 exosomal circulating miRNAs differently expressed between somatotroph adenomas and healthy pituitary samples during the serum analysis of patients with somatotroph adenomas [53]. Among the 169 miRNAs, miR-423-5p was expressed lower in somatotroph adenomas than in healthy pituitary samples, which was proven by miRSCan Panel Chip qPCR. Circulating miRNAs derived from exosomes may serve as diagnostic and prognostic biomarkers by identifying specific RNA signatures of PA cells. In addition, miR-423-5p induced cell apoptosis, inhibited cell proliferation, and reduced growth hormone release and migration of GH3 cells. Exosomal circulating miR-423-5p may act as biomarkers of somatotroph adenomas, but its application as a biomarker needs further investigation.

Kelly et al., by utilizing two separate methods, namely, microarrays and qRT-PCR, identified four candidate circulating miRNAs (miR-663, miR-2861, miR-3152, and miR-3185) as biomarkers of recombinant human GH (rhGH) used in subjects with and without rhGH administration [54]. The authors identified and confirmed four circulating miRNAs that were differentially expressed in all individuals using therapeutic replacement doses of rhGH when compared to individuals with naturally high levels of GH and normal controls. This study further develops the hypothesis that circulating miRNAs may be used as biomarkers for the detection of gonadotropin-secreting adenomas, but it remains to be seen.

Using next-generation sequencing (NGS), Lutsenko et al. performed profiling on four circulating miRNAs (miR-4446-3p, miR-215-5p, miR-342-5p, and miR-191-5p) in the plasma of 12 patients with acromegaly compared with 12 healthy controls [55]. They found decreased expression of these four circulating miRNAs. The aim of this study was to identify changes in the expression of circulating miRNAs in patients with acromegaly compared with healthy control. However, the results must be confirmed using various measurement methods, such as qRT-PCR with a large sample size, for potential consideration of these circulating miRNAs as biomarkers.

PA secreting prolactin may also suppress ovulation. In particular, microadenomas often produce elevated levels of prolactin, which causes anovulation. Eisenberg et al. suggest that miR-200b and miR-429 are involved in ovarian function via a mechanism of pituitary control in vitro [56]. In addition, their study identified circulating miR-200b and miR-429 in human serum with a temporal downward change during the normal ovulatory cycle. These circulating miRNAs were overexpressed in ovulatory women, and their high levels drastically dropped only after administration of exogenous gonadotropins. Chen et al. found that miR-200b involved in the physiological process of development, growth, and hormone secretion of prolactinoma [57]. In another study, the expression of miR-200b and protein kinase Cα (PKCα) in pituitary tumors was investigated to determine whether miR-200b may inhibit proliferation and invasion of pituitary tumor cells [58]. The involvement of miR-429 in humans has been tied to tumorigenesis, particularly in the epithelial-to-mesenchymal transition (EMT) [59]. miR-132 and miR-15A/16 overexpression facilitate the impairment of PA tumor cell proliferation and migration and blocks the proteins involved in EMT. Both of these miRNAs particularly could target Sox5 in a synergistic manner and induce blockade of invasive PA tumor cell proliferation. Hence, the development of miRNA-based therapeutic targets to block pituitary tumor cells proliferation could be a significant approach to improve clinical outcome [27].

## 4. miRNAs and PA Therapy

Regarding the use of miRNAs as therapeutic agents in tumors, including PA, there are two main strategies: (1) restoration of miRNA suppressors of oncogenes, the expression of which is suppressed in the tumor; or (2) inhibition of overexpressed “oncomiRs” [60]. The restoration of tumor suppressor miRNA can be achieved using miRNA mimics, which are synthetic double-stranded RNA molecules with the same sequence as target miRNAs that are able to integrate into RNA, inducing RNA- induced silent complex (RISC), and perform an antitumor function. Anti-miRNA therapy is aimed at suppressing the expression of “oncomiRs” that are overexpressed in the tumor. Recently, there have been many strategies aimed at suppressing this expression, which can potentially be introduced into clinical practice. Inhibition of such miRNAs can be achieved by antagomiRs (oligonucleotides capable of suppressing “oncomiRs” conjugated to cholesterol to facilitate cell uptake), antisense oligonucleotides (AMOs), miRNA masks (modified 2’-O-methyloligonucleotide, complementary to the binding sites of miRNAs with mRNA targets), blocked nucleic acid (oligonucleotides with an LNA ribose fragment modified to improve specificity and stability), low molecular weight miRNA inhibitors (SMIRs), and miRNA sponge [60,61].

miRNA-based therapeutic modalities to foster the inhibition of vascular endothelial growth factor (VEGF) and mTOR as well as the regulation of tyrosine kinase inhibitors (TKIs) and somatostatin receptors (SSTRs) may be beneficial for future studies as an effective therapy for benign and aggressive PAs. The VEGF family is a factor in vascular permeability, and it plays a significant role in tumor angiogenesis, stimulating the proliferation and migration of endothelial cells (ECs) [62]. The extensive expression of VEGF in invasive and/or aggressive tumors has been previously reported [62]. In addition, several studies have reported a correlation between the hormonal profile of PA and the overexpression of VEGF [63]. There is also the suggestion that VEGF can increase cell survival by inducing the expression of the antiapoptotic B-cell lymphoma 2 (BCL-2) proteins in PA [64]. Because miR-126 is specific for ECs, performing the function of preserving vascular integrity and angiogenesis, it is a potential target for effective anti-miRNA therapy in situations of aberrant vascularization, including PA [65]. miR-132 was overexpressed in tumor ECs, and hemangiomas was not detected in normal endothelium [66]. Studies have shown the systemic administration of anti-miR-296 and anti-miRNA-132 with the use of cRGD-modified nanoparticles to inhibit tumor angiogenesis in vivo, which opens up prospects for miRNA-based antitumor therapy for PA [67]. The epidermal growth factor receptor (EGFR) has attracted interest as a potential therapeutic target for resistant and aggressive PA, mainly prolactinomas and corticotroph adenoma [68]. TKIs block the cascades of EGFR signaling and, in primary human prolactinoma, cultures reduce prolactin levels [69]. It is known that the use of miRNA mimics, such as miR-7, miR-128, miR133b, miR-146a, miR-302b, and miR-608, regulate the expression of receptor tyrosine kinase (RTK) in solid tumors, inhibiting the expression of EGFR in vitro [70].

mTOR is a serine–threonine specificity protein kinase that controls cell growth and metabolism in response to various substances, growth factors, and cell stress [71]. As a central regulator of cell growth, mTOR plays a key role in the development and aging of cells and is involved in the pathogenesis of cardiovascular diseases, endocrine diseases, metabolic disorders, and tumors [72]. Antiproliferative responses to mTOR inhibition have been reported in studies with aggressive PA in vitro and in vivo [73,74]. mTOR mRNA expressions significantly modulated in invasive pituitary adenoma tissues were compared with those in noninvasive pituitary adenoma tissues [73,74].

Some miRNAs have been shown to modulate multidrug resistance in many tumor cells, including PAs. For instance, Wu et al. identified that miR-93 was significantly upregulated in bromocriptine-resistant prolactinomas [75]. In addition, the silencing of mir-93 significantly increased the sensitivity of MMQ cells to dopamine agonist treatment. Their miRNA–mRNA network analysis predicted that miR-93 could potentially regulate p21 expression. In another study, Hu et al. confirmed that miR-93-5p could enhance the drug resistance of prolactinoma cells by regulation of TGF-β1/Smad3-dependent fibrosis [76]. Jian et al. explored the relationship between miR-145-5p expression as well as bromocriptine sensitivity both in vitro and in vivo [77]. They demonstrated that miR-145-5p was decreased in bromocriptine-resistant prolactinoma tissues; the downregulation of miR-145-5p was correlated with the upregulation of translationally controlled tumor protein (TPT1) in this study. Taken together, this study suggested that miR-145-5p affected the sensitivity of prolactinoma to bromocriptine by directly binding to the 3′-UTR of TPT1. miR-145-5p might be a promising candidate target to develop a novel therapeutic strategy to overcome the bromocriptine resistance of prolactinoma.

Different somatostatin receptors (SSTRs) are expressed in many endocrine tumor tissues, such as PAs. Five subtypes of SSTRs, including SSTR1-5, SSTR2, and SSTR5 expression, were observed in GH-secreting PA [78,79]. Somatostatin analogs (SSAs), including lanreotide and octreotide, are most widely used to treat GH-secreting PA [79]. Although SSAs are associated with high symptomatic response rates initially, patients with PA may develop resistance to treatment over time. It is likely that some differentially expressed miRNAs may affect a patient’s response to SSAs. For instance, using microarray analysis, Mao et al. found a differential expression of miRNAs between GH-secreting PAs and normal pituitary and SSA responding and nonresponding patients [80]. Furthermore, qRT-PCR results for the expression of miR-124, miR-125a, miR-126, miR-223, miR-381, miR-503, miR-524-5p, miR-525-5p, and miR-886-5p were consistent with the results obtained from microarray analysis. Further analyses showed that these miRNAs are associated with pituitary tumor-transforming gene (PTTG), insulin-like growth factor-binding protein 3, 7 (IGFBP-3, 7), and insulin-like growth factor-binding protein complex acid labile subunit chain precursor (IGFALS), which are involved in protein binding, receptor binding, cell communication, and regulation of growth. Their results indicate that altered miRNA expression is involved in GH-secreting PA transformation, which will shed light on the mechanisms for the treatment of acromegaly by SSAs. Further research showed that miR-185 targeted SSTR2 mRNA to downregulate SSTR2 protein expression, promote proliferation, and inhibit apoptosis of tumor cells, which would be classified as an oncogene in GH-secreting PA cells [81]. Differential expression of miR-185 was also observed in SSA responder and nonresponder GH-secreting PA. Their study demonstrated that miR-185 is likely involved in drug resistance and the pathogenesis of GH-secreting PAs. However, further research is needed to elucidate the regulation mechanism by miR-185 in PAs, particularly GH-secreting PA. In another study, miR-7 and miR-148a that were highly induced by SST analogs were shown to inhibit the proliferation of NCI-H727 and CNDT2 cells. SST analogs also produced a general upregulation of the let-7 family members. SST analogs control and induce distinct miRNA expression patterns, among which miR-7 and miR-148a both have growth inhibitory properties. In PA, the role of miRNA in drug resistance is an attractive area of research and is expected to lead to the development of novel miRNA-based therapies in the future.

## 5. Challenges of miRNAs as Diagnostic/Therapeutic Tools

Extensive research has been conducted to apply miRNAs as routine diagnostics for PAs. However, the challenging preanalytics and disparities in exosomal RNA isolation may affect the use of circulating RNAs as biomarkers to diagnose PA. The ability of certain miRNAs to target both tumor suppressor genes and oncogenes is conducive to the duplicity of clinical responses further making hard to the oncologists to predict many PAs-related malignancies [82].

The isolated exosomes from different cells often contain “contaminants” accompanied by the heterogeneity. The removal process of these contaminants is difficult due to the disadvantages pertaining to the selection and reproduction of different isolation methods [83]. In addition, the abundance of coisolated proteins, viz. albumin, immunoglobulins, matrix metalloproteinases (MMPs), and other immune components in body fluids make it hard for oncologists to use miRNAs as prognostic tools for PAs. The isolation and detection of exosomes through immunocapture methods using specific binding of the detection antibody to an antigen on the exosome surface might be beneficial to eliminate “contaminants” [83]. However, the available research data delineate the hurdles in implementing suitable exosome isolation approaches from plasma or body fluids of patients with PAs due to the presence of the above components [84,85]. Hence, intense research is required to design novel methodologies to perform appropriate analysis on the isolation of exosomes for selected miRNAs with potential diagnostic and prognostic value for PAs.

miRNAs, as post-transcriptional regulators, play an important role in the regulation of many cellular functions and can therefore also play a role in the development of PA. There is an assumption based on numerous studies that diagnostic and prognostic analysis can be established based on the expression profile of circulating miRNAs in biological fluids, such as blood, in various diseases, including tumors. To date, there are few reports on the study of circulating miRNAs in patients with PA. Profiling of the expression of circulating miRNAs can potentially be used to assess relapse and follow-up and study the effectiveness of chemo and radiation therapy for PAs with aggressive behavior, such as invasive PAs. In addition, modern methods of treating aggressive pituitary adenomas are not completely effective, and miRNAs are currently considered as new therapeutic agents designed for specific signaling pathways involved in the tumor process. All this opens up new opportunities for the study of miRNAs as potential new biomarkers and therapeutic agents for PA.

## Figures and Tables

**Figure 1 ijms-21-07287-f001:**
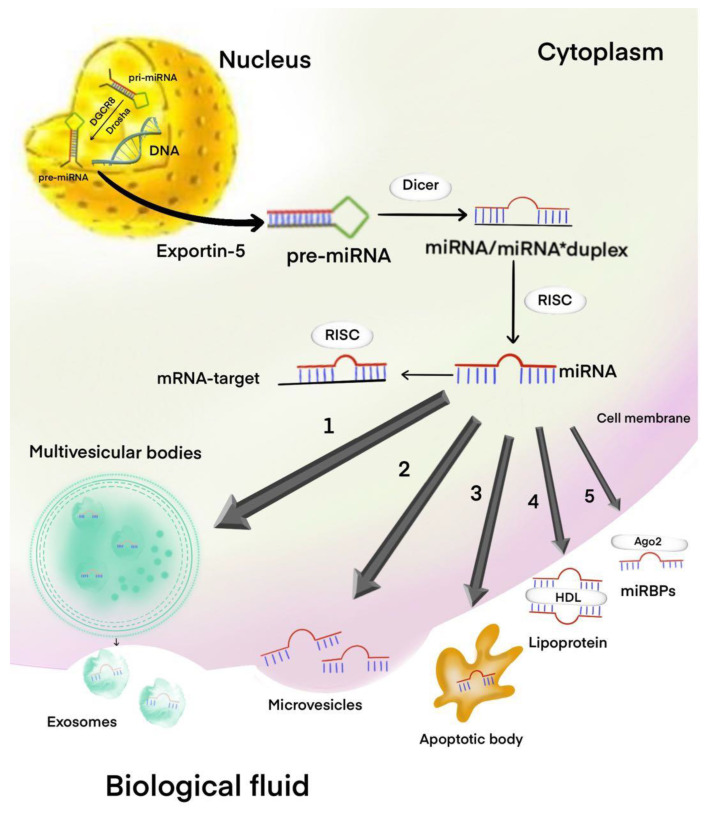
MicroRNA (miRNA) biogenesis and release of miRNAs in the extracellular environment. Pri-miRNA processed by Drosha/DGCR8 to pre-miRNA. Exportin-5 transfers these pre-miRNAs from nucleus to cytoplasm, where the dicer processes them into mature miRNAs. Mature miRNAs can be selectively incorporated into the exosomes (**1**), microvesicles (**2**), apoptotic body (**3**), high-density lipoprotein (HDL) (**4**), or coupled with miRNA and RNA-binding proteins (miRBPs) like Ago2 protein (**5**) and released in to biological fluid.

**Table 1 ijms-21-07287-t001:** Experimental information on miRNAs involved in the tumorigenesis of pituitary adenomas and their target genes by tumor type.

miRNA	Gene Target	Type of PA	Biological Function	Regulation	Phenotype	Ref.
miR-524-5p	PTTG1/PBF	NFA	Inhibition of tumor cell proliferation, migration, and invasion	Down	Tumor suppressor	[19]
miR-424, miR-503	CDC25A	NFA	Inhibition of tumor cell growth	Down	Tumor suppressor	[20]
miR-34	AIP	GH	Induces invasive properties in tumor cells	Up	OncomiR	[21]
miR-410	CCNB1	FSH/LH	Increased expression of cyclin A and D protein affecting the G1-S phase of the cell cycle; inhibition of tumor cell proliferation	Down	Tumor suppressor	[22]
miR-26a	PLAG1	NFA, GH, ACTH, PRL	Induces invasive properties in tumor cells	Up	OncomiR	[23]
miR-23b	HMGA2	NFA, FSH/LH, GH	Inhibition of tumor cell proliferation, delaying cell division in the G1 phase of the cell cycle	Down	Tumor suppressor	[24]
miR-130b	CCNA2	NFA, FSH/LH, GH	Inhibition of tumor cell proliferation, delaying cell division in the G2 phase of the cell cycle	Down	Tumor suppressor	[24]
miR-106b	PTEN	Invasive (NFA, GH, ACTH, PRL)	Induces invasive properties in tumor cells	Up	OncomiR	[25]
MIR-376B-3P with MEG3	HMGA2	Clinical nonfunctioning pituitary adenomas (CNFPAs)	HMGA2, acting as a target gene of MIR-376B-3P, functions as an oncogene in PA and can be downregulated by MEG3 through the accumulation of MIR-376B-3P	-	Tumor suppressor	[26]
miR-132, miR-15a, miR-16	SOX5	Invasive (NFA, GH, ACTH, PRL)	Inhibition of tumor cell proliferation, migration, and invasion	Down	Tumor suppressor	[27]
miR-200c	PTEN	PRL	Tumor cell apoptosis reduction	Up	OncomiR	[28]
miR-329, miR-300, miR-381, miR-655	PTTG1	PRL, GH	Inhibition of tumor cell growth	Down	Tumor suppressor	[29]
miR-20a, miR-17-5p	PTEN, TIMP2	NFA	Carcinoma metastasis	Up	OncomiR	[27]
miR-106b	PTEN-PI3K/AKT	NFA	Activation of invasive properties and migration of tumor cells; carcinoma metastasis	Up	OncomiR	[27,30]
miR-183	KIAA0101	PRL	Inhibition of tumor cell proliferation	Down	Tumor suppressor	[31]
miR-15, miR-16, miR-26a, miR-196a2, Let-7a	HMGA1, HMGA2	GH, PRL	Inhibition of tumor cell proliferation	Up	Tumor suppressor	[32]

NFA, nonfunctioning adenoma; GH, growth hormone-secreting adenoma; FSH, follicle-stimulating hormone adenoma; LH, luteinizing hormone-secreting adenoma; ACTH, adrenocorticotropic hormone-secreting adenoma; PRL, prolactin-secreting adenoma; PTTG1IP, pituitary tumor-transforming 1 interacting protein; PBF, pituitary tumor-transforming gene binding factor; CDC25A, cell division cycle 25 A; AIP, aryl hydrocarbon receptor-interacting protein; CCNB1, G2/mitotic-specific cyclin-B1; PLAG1, pleomorphic adenoma gene 1; HMGA1, high-mobility group AT-hook 1; HMGA2, high-mobility group AT-hook 2; CCNA2, cyclin A2; PTEN, phosphatase and tensin homolog deleted on chromosome 10; AKT3, serine/threonine kinase 3; SOX5, SRY-related HMG-box; SSTR2, somatostatin receptor type 2; TIMP2, tissue inhibitor of metalloproteinases 2; PI3K, phosphoinositide 3-kinases; KIAA0101, protein; miR, microRNA; PA, pituitary adenoma.

**Table 2 ijms-21-07287-t002:** Circulating miRNAs in Preoperative Endocrine Tumors

miRNA.	Tumor Type	Sample	Comparison	Sensitivity %	Specificity %	AUC	Regulation	Ref.
Exosomal miR-101 and miR-483-5p	ACA and ACC	Plasma	ACC vs. ACA	68.75 and 87.5	83.33 and 94.44	0.766 and 0.965	Up	[35]
miR-124-3p, miR-9-3p and miR-196b-5p	PTC and benign thyroid nodules	Plasma	PTC vs. benign lesion/PTC vs. healthy control	88, 80, and 74	78.8, 73.7, and 66	0.859, 0.823, and 0.781	Up	[36]
miR-25-3p and miR-451a	PTC and benign thyroid nodules	Plasma	PTC vs. benign lesion	92.8 and 88.9	68.8 and 66.7	0.835 and 0.857	Up	[37]
miR-34a and miR-483-5p	ACA and ACC	Serum	ACC vs. ACA	/	/	0.81 and 0.74	Up	[38]
miR-483-5p	ACA and ACC	Plasma	ACC vs. ACA	87	78.3	0.88	Up	[39]

ACA, adrenocortical adenoma; ACC, adrenocortical carcinoma; PTC, papillary thyroid carcinoma; miR, microRNA; AUC, area under the receiver operating characteristic (ROC) curve.

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
