# Peer review of "MiRNAs as Noninvasive Biomarkers and Therapeutic Agents of Pituitary Adenomas"

_ijms, 2020, doi:10.3390/ijms21197287_

Round 1
Reviewer 1 Report
The authors performed a systematic review to interpret miRNAs’ role in pituitary adenomas. From their report, they found several abnormally expressed miRNAs and their potential mechanisms in pituitary adenomas. This is an interesting review, however, they just used two tables to list these miRNAs and related subsequent data to introduce these miRNAs and their roles. I think this kind of presentation may bring difficulty to readers and the readers may need spend more time to get the main idea of this paper. Thus, if they give us only one more figure to present these miRNAs and related roles in pituitary adenomas, it may make the paper more readable.
Author Response
Dear Reviewer 1,
On behalf of my coauthors, please accept my sincere thanks and gratitude for careful perusal and critical review of our manuscript entitled “MiRNAs as non-invasive biomarkers and therapeutic agents of pituitary adenomas”. We have revised the manuscript based upon the reviewers’ comments as well as your suggestion. Adequate care has been taken to accommodate each and every suggestion of the reviewers. An itemized, “point-by-point” reply to all the comments is attached separately where we have clearly presented our specific response and additions, deletions and/or modifications that have been made in the revised text, and highlighted.
Point-by-Point Answers to the Comments
Reviewer-1
Comment-1: This is an interesting review, however, they just used two tables to list these miRNAs and related subsequent data to introduce these miRNAs and their roles. I think this kind of presentation may bring difficulty to readers and the readers may need spend more time to get the main idea of this paper. Thus, if they give us only one more figure to present these miRNAs and related roles in pituitary adenomas, it may make the paper more readable.
Response: Authors thank the reviewer’s suggestion and we have described the role of each miRNA appropriately where figure requirement is not necessary.
Reviewer 2 Report
The authors have outlined the miRNA role as non invasive biomarkers and therapeutic agents of pituitary adenomas.
The manuscript provides a succinct but very informative and comprehensive status of miRNA role in pituitary adenomas.
Overall, the data are well collected into each table and the conclusions are convincing.
Author Response
Dear Reviewer 2,
On behalf of my coauthors, please accept my sincere thanks and gratitude for careful perusal and critical review of our manuscript entitled “MiRNAs as non-invasive biomarkers and therapeutic agents of pituitary adenomas”. We have revised the manuscript based upon the reviewers’ comments as well as your suggestion. Adequate care has been taken to accommodate each and every suggestion of the reviewers. An itemized, “point-by-point” reply to all the comments is attached separately where we have clearly presented our specific response and additions, deletions and/or modifications that have been made in the revised text, and highlighted.
Reviewer 2
Comment-1: The authors have outlined the miRNA role as non invasive biomarkers and therapeutic agents of pituitary adenomas. The manuscript provides a succinct but very informative and comprehensive status of miRNA role in pituitary adenomas. Overall, the data are well collected into each table and the conclusions are convincing.
Response: Authors thanks the reviewer.
Point-by-Point Answers to the Comments
Reviewer 3 Report
The manuscript by Beylerli O. et.al. offers an interesting overview on the role of microRNAs as diagnostic/prognostic/therapeutic tools in pituitary adenoma. The manuscript focuses on the expression levels of the miRNAs in blood, as future minimally invasive ways to assess diagnostic and therapy response in pituitary adenomas. Still, there are some issues that the authors need to address.
Major issue:
- The discussion throughout the text seems too much one-sided, little information is provided on the challenges/limitations when it comes to the use of miRNAs as diagnostic/therapeutic tools in cancer. It would be useful to add 1-2 paragraphs at the end of two chapters: “Circulating miRNAs and PA” and “MiRNAs and PA therapy” that should contain more details on the disadvantages/challenges faces by the use of miRNAs as diagnostic/therapeutic tools should be address, such as the off target effects, the difference in exosomal RNA isolation and analysis methods that may affect the use of circulating RNAs as biomarkers, the ability of some miRNAs to target both tumor suppressor and oncogenes thus having a duplicity and hard to predict response in many malignancies and others
Minor issues:
-Line 45-46: please rephrase the following sentence: “However, in many biological fluids of the human body, like blood, numerous miRNAs, called circulating, have been detected.”
-Line 67, please correct the word “mandala”
-Line 71-72, please rephrase the following sentence: “The content of exosomes is very diverse, including miRNAs.”
-Line 79, please correct “oncovirus”, the correct form is “oncomiRs”
-Line 83, please provide two more examples of pathways affected by miRNAs, besides MYC.
-Line 86, replace “oncomes” with “oncomiRs”
-Line 87, please delete :” This gives an idea of the study of specific pathogenesis, diagnosis and treatment of PA.”
-Figure 1, add an arrow in the nucleus from the DNA to pri-miRNA and move the Exportin-5 arrow a little below
-line 91, correct “prim-miRNA” and provide the full name
-line 95, correct “miRNA-binding proteins (mobs) ”, with “microRNA (miRNA) and RNA-binding proteins (RBPs) (abbreviated miRBPs)”
-lines 104-105, “role of miRNAs in pathogenesis different types of PAs”, correct to “role of miRNAs in pathogenesis of different types of PAs”
-lines 110-112: For the following phrase “Some miRNAs may be directly involved in PA development by controlling cell differentiation and apoptosis, while others may be involved in PAs by targeting oncogenes and/or tumor suppressor genes.”, please clarify if you are referring here to miRNAs that on one hand have a role in biological processes and on the other hand, have a role in molecular changes
-Table 1, the reference 24 and all the information provided in that row is the same as the one provided for ref.15; also, the reference 25 is wrong, the information provided in that row is found in reference 26.
-line 138: “Because early detection of tumors in the early stages”, please delete the first “early”
-line 178, it is not “morbidity and mortality even”, but “morbidity and mortality events”
-line 232, please add “of”, in “in plasma 12 patients”
-line 249, ref.55 refers to lung cancer, can you provide at least one additional example in brain tumors?
-line 260, replace “miRNA suppressor of oncogenes”, with “tumor suppressor miRNA”
-line 273, contains an odd phrasing referring to miRNA targets: ” in the face of miRNAs”, please rephrase
-line 307, correct “mir-93”, with “miR-93”
-line 317, please provide the full name of SSTRs
Author Response
Dear Reviewer 3,
On behalf of my coauthors, please accept my sincere thanks and gratitude for careful perusal and critical review of our manuscript entitled “MiRNAs as non-invasive biomarkers and therapeutic agents of pituitary adenomas”. We have revised the manuscript based upon the reviewers’ comments as well as your suggestion. Adequate care has been taken to accommodate each and every suggestion of the reviewers. An itemized, “point-by-point” reply to all the comments is attached separately where we have clearly presented our specific response and additions, deletions and/or modifications that have been made in the revised text, and highlighted.
Reviewer 3
Major issues
Comment-1: The discussion throughout the text seems too much one-sided, little information is provided on the challenges/limitations when it comes to the use of miRNAs as diagnostic/therapeutic tools in cancer. It would be useful to add 1-2 paragraphs at the end of two chapters: “Circulating miRNAs and PA” and “MiRNAs and PA therapy” that should contain more details on the disadvantages/challenges faces by the use of miRNAs as diagnostic/therapeutic tools should be address, such as the off target effects, the difference in exosomal RNA isolation and analysis methods that may affect the use of circulating RNAs as biomarkers, the ability of some miRNAs to target both tumor suppressor and oncogenes thus having a duplicity and hard to predict response in many malignancies and others
Response: Authors included the suggested paragraphs and highlighted in green.
Minor issues
Comment-2: Line 45-46: please rephrase the following sentence: “However, in many biological fluids of the human body, like blood, numerous miRNAs, called circulating, have been detected.”
Response: Authors corrected the text.
Comment-3: -Line 67, please correct the word “mandala”
Response: Authors corrected the text.
Comment-4: -Line 71-72, please rephrase the following sentence: “The content of exosomes is very diverse, including miRNAs.”
Response: Authors corrected the text.
Comment-5: -Line 79, please correct “oncovirus”, the correct form is “oncomiRs”
Response: Authors corrected the text.
Comment-6: -Line 83, please provide two more examples of pathways affected by miRNAs, besides MYC.
Response: Authors corrected the text.
Comment-7: -Line 86, replace “oncomes” with “oncomiRs”
Response: Authors corrected the text.
Comment-8: -Line 87, please delete :” This gives an idea of the study of specific pathogenesis, diagnosis and treatment of PA.”
Response: Authors corrected the text.
Comment-9: -Figure 1, add an arrow in the nucleus from the DNA to pri-miRNA and move the Exportin-5 arrow a little below
Response: Authors corrected the text.
Comment-10: -line 91, correct “prim-miRNA” and provide the full name
Response: Authors corrected the text.
Comment-11: -line 95, correct “miRNA-binding proteins (mobs) ”, with “microRNA (miRNA) and RNA-binding proteins (RBPs) (abbreviated miRBPs)”
Response: Authors corrected the text.
Comment-12: -lines 104-105, “role of miRNAs in pathogenesis different types of PAs”, correct to “role of miRNAs in pathogenesis of different types of PAs”
Response: Authors corrected the text.
Comment-13: -lines 110-112: For the following phrase “Some miRNAs may be directly involved in PA development by controlling cell differentiation and apoptosis, while others may be involved in PAs by targeting oncogenes and/or tumor suppressor genes.”, please clarify if you are referring here to miRNAs that on one hand have a role in biological processes and on the other hand, have a role in molecular changes
Response: Authors corrected the text.
Comment-14: -Table 1, the reference 24 and all the information provided in that row is the same as the one provided for ref.15; also, the reference 25 is wrong, the information provided in that row is found in reference 26.
Response: Authors corrected the text.
Comment-15: -line 138: “Because early detection of tumors in the early stages”, please delete the first “early”
Response: Authors corrected the text.
Comment-16: -line 178, it is not “morbidity and mortality even”, but “morbidity and mortality events”
Response: Authors corrected the text.
Comment-17: -line 232, please add “of”, in “in plasma 12 patients”
Response: Authors corrected the text.
Comment-18: -line 249, ref.55 refers to lung cancer, can you provide at least one additional example in brain tumors?
Response: Authors corrected the text.
Comment-19: -line 260, replace “miRNA suppressor of oncogenes”, with “tumor suppressor miRNA”
Response: Authors corrected the text.
Comment-20: -line 273, contains an odd phrasing referring to miRNA targets: ” in the face of miRNAs”, please rephrase
Response: Authors corrected the text.
Comment-21: -line 307, correct “mir-93”, with “miR-93”
Response: Authors corrected the text.
Comment-22: -line 317, please provide the full name of SSTRs
Response: Authors corrected the text.